# DeCo: Decoupling Token Compression from Semantic Abstraction in Multimodal Large Language Models

## Abstract

The visual projector, which bridges the vision and language modalities and facilitates cross-modal alignment, serves as a crucial component in Multimodal Large Language Models (MLLMs). However, measuring the effectiveness of projectors in vision-language alignment remains under-explored, with current evaluations relying primarily on the performance of MLLMs on downstream tasks. Motivated by this gap, this study conducts an in-depth examination of the projector module by analyzing the vision-language semantic flow within MLLMs. Our findings reveal that compressive projectors (e.g., QFormer) reduce the number of visual tokens by abstracting visual patches into a limited set of semantic concepts, such as objects or attributes, leading to a deficiency we term "double abstraction" in MLLMs. This phenomenon involves i) an initial visual semantic abstraction by the projector in the vision modality, which refers to pre-defined query tokens, and ii) a secondary extraction by the LLM in the language modality based on text instructions. The double abstraction is inefficient during training and leads to cumulative deficiencies in visual semantics. To address this issue, we propose the key insight of "**De**couple Token **Co**mpression from Semantic Abstraction (**DeCo**)", where projectors compress visual tokens at the patch level non-semantically, while allowing the LLM to fully manage semantic understanding and abstraction. Consequently, we employ a simple compressor, i.e., 2D Adaptive Pooling, to downsample visual patches in a parameter-free manner. Empirical evaluations demonstrate that 2D Adaptive Pooling outperforms traditional compressive projectors in both performance and efficiency, achieving gains of 0.9%, 7.1%, and 2.9% across the MLLM Benchmarks, Visual Localization, and Open-ended VQA tasks, respectively, while utilizing fewer trainable parameters and achieving faster convergence. Furthermore, it preserves vision spatial locality and exhibits robustness across various MLLM configurations, including different vision backbones, image resolutions, and LLMs.

## 1 Introduction

Multimodal Large Language Models (MLLMs) (OpenAI, 2023; Gemini Team, 2023; Reka, 2024) endow Large Language Models (LLMs) with vision perception capability, which have shown their versatility and expertise in diverse vision-language tasks (Kafle et al., 2018; Yu et al., 2016; Singh et al., 2019; Bigham et al., 2010; Li et al., 2024; Yao et al., 2023; 2022; Chen et al., 2023b). For MLLMs, learning good vision-language alignment is at the core of their intelligence (Li et al., 2023d; Zhu et al., 2023; Ren et al., 2023b; 2024). To achieve cross-modal alignment, recent studies utilize an intermediate module, i.e., the projector (Liu et al., 2023b; Zhu et al., 2023; Madureira, 2021; Dai et al., 2023), to map representations of image patches (Dosovitskiy et al., 2020) into the LLM embedding space as visual tokens.

Widely used projectors can be roughly summarized into two branches: non-compressive and compressive. The non-compressed projector (Liu et al., 2023b) directly uses linear layers that translate the visual token dimension to the LLM's while keeping the visual token number unchanged. Despite its simplicity and effectiveness, the linear projector struggles with high training resources and costs due to the length of the visual token sequence. The sequence would be long in two common scenarios: (i) the length increases quadratically with the input resolution (Li et al., 2023a; Chen et al., 2023c); (ii) the length increases linearly with the image number for handling video frames (Ren et al., 2023c;

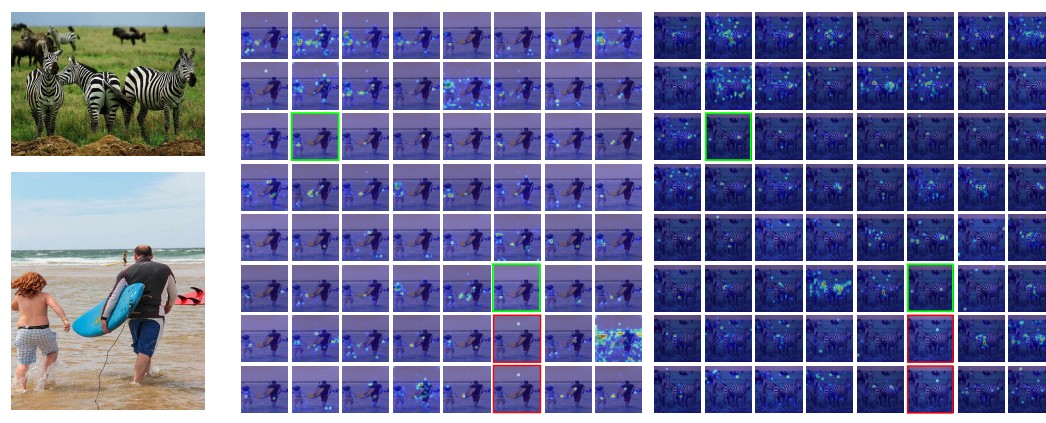

(a) Original Images  (b) Query-to-Patch Relevance (down img)  (c) Query-to-Patch Relevance (top img)

Figure 1: Visualization of the R-GAE relevance map from compressed visual tokens (Query) to original image patches (Patch) of the QFormer (Li et al., 2023d) projector. The QFormer reduces the original 576 visual (patch) tokens to 64 (equal to $8 \times 8$) learned query tokens. The relevance maps are obtained from the image-to-text generation process of the MLLM. From the Query-to-Patch map (zoomed in), each query token is activated with diverse visual concepts at the semantic level, such as objects (zebras, grassland, the skateboard), attributes (black and white texture of zebras), and backgrounds (the sea level). However, different query tokens from the same image are visually sparse and showcase repetitive patterns (highlighted in the same color frame), limiting their capacity for visual semantic expression.

Song et al., 2023; Ren et al., 2023a), potentially resulting in sequences up to a million tokens long (Liu et al., 2024a). On the other branch, prevalent compressive projectors, e.g., QFormer (Li et al., 2023d; Dai et al., 2023), Resampler (Alayrac et al., 2022), and D-Abstractor (Cha et al., 2024), condense the original visual tokens into fewer query tokens to reduce visual redundancy, which have a better balance between performance and efficiency.

However, how existing projectors affect the vision-to-language semantic alignment in an explainable perspective is still under-explored. Understanding this question is crucial for facilitating better architectural improvement and providing broader practicability in demanding scenarios such as high image resolutions and video applications. In this study, we investigate this problem by analyzing the relevance between generated textual tokens, raw visual patches and intermediate projector outputs. We start by tracing the language-to-vision semantic flow using a novel R-GAE explainability tool. Specifically, we decouple the overall Text-to-Patch semantic relevance to Text-to-Query and Query-to-Patch sub-flows during the image-to-text generation. Among the sub-flows, the Text-to-Patch relevance reveals the effective visual context from ViT patch tokens (Patch) leveraged by the LLM (Text). Meanwhile, the Query-to-Patch relevance interprets the visual patterns learned from original visual patches (Patch) by query tokens (Query).

Based on the R-GAE analysis, we derive two important findings: **Firstly**, the query tokens *compress* the number of visual tokens by *abstracting* semantic-level visual concepts, leading to visual semantics deficiency such as loss of fine-grained attributes and spatial locality. As Figure 1 illustrates, different query tokens are activated with varied visual concepts such as objects, attributes or backgrounds from the original images. For the top image with zebras in the grassland, query tokens attend to visual patterns such as three zebras, their body parts, surface textures, and distant backgrounds respectively. However, the fixed number of queries can only express limited visual semantics. Specifically, different query tokens show repetitive patterns across images (highlighted by color frames in Figure 1). Moreover, they tend to lose fine-grained visual attributes (e.g., "purple and red" in Figure 3). Furthermore, the vanilla QFormer has been demonstrated to lose visual spatial locality (Cha et al., 2024) during semantic abstraction.

**Secondly**, the LLM acts as an excellent visual-semantic abstractor directly from patch features. As Figure 3 first row shows, utilizing a non-compressive linear projector allows the LLM to perceive patch-level visual representations and attend to accurate vision regions without prior semantic deficiency. Consequently, the QFormer-based MLLM system redundantly extracts visual semantics twice—once by the QFormer and again by the LLM—a phenomenon we refer to as *Redundant Double Abstraction*. This double abstraction introduces two major drawbacks: (i) increased training resource,

Figure 2: The overall analysis framework of a typical MLLM. During image-to-text generation, we trace back the language-to-vision semantic flow utilizing R-GAE relevance maps.

such as GPUs and data, are required to optimize an external visual semantic abstractor as the projector, and (ii) without careful training, an accumulation of visual semantic deficiencies will propagate from the projector to the LLM, such as the loss of fine-grained semantics and spatial locality caused by a poorly performing QFormer. As a result, the initial visual semantic abstraction by QFormer adds unnecessary burden to the MLLM system.

To overcome the double abstraction problem, we propose to **De**couple token number **Co**mpression (**DeCo**) from vision semantic abstraction. The core of DeCo is using *a simpler projector, which operates and outcomes visual tokens directly at the patch level non-semantically to reduce the visual token number*. Subsequently, the LLM acts as an expertise to understand and abstract both visual and textual semantics. To quantitatively validate the DeCo insight, we adopt the naive Adaptive Average Pooling as a natural down-sampler at the patch level and then use the linear layers to map the reduced visual tokens. Under fair experimental settings, quantitative results demonstrate that a simple Adaptive Pooling design consistently outperforms semantic-level compressed projectors in both effectiveness and efficiency. Additionally, experiments across various MLLM configurations, including different vision backbones, image resolutions, and LLMs, further highlight the robustness of Adaptive Pooling. Through both qualitative and quantitative analysis, our DeCo insight aims to illuminate ways to improve the efficiency of the projector module in current MLLM systems. We also hope that it will serve as a valuable reference for future architectural improvements in projector design.

## 2 VISUAL PROJECTOR ANALYSIS

In this section, we analyze the impact of projector modules in Multimodal Large Language Models (MLLMs) from a semantic flow perspective using a novel R-GAE explainability tool. During image-to-text generation, visual context plays an indispensable role in the perception of Large Language Models (LLMs). The related relevance maps between image and text, such as attention maps (Vaswani et al., 2017), can serve as an interpretation of the vision-language semantic alignment (Chefer et al., 2021b; Xu et al., 2015; Carion et al., 2020; Ren et al., 2021). As Figure 2 shows, given an oracle description in the MLLM architecture, the backtracking relevance map from text words to visual patches (referred to as Text-to-Patch) exhibits the visual semantics aligned with the LLM and further indicates the effective visual context leveraged by the LLM. To examine the impact of projectors as the intermediate module, we dissect the Text-to-Patch relevance map into Text-to-Query and Query-to-Patch sub-maps. The Query-to-Patch map can explain the visual patterns learned by the query (or compressed) tokens, while the difference between Text-to-Patch and Text-to-Query, exerted by the projector, reveals its impact on the vision-language semantic alignment.

### 2.1 PROBLEM FORMULATION

A typical MLLM architecture comprises a Vision Transformer (ViT) to acquire visual representations $\mathcal{I} \in \mathbb{R}^{N \times d_I}$ containing $N$ patches, a projector to transform visual representations into the textual embedding space, and an LLM that handles both vision and instruction tokens to output hidden states $\mathcal{T} \in \mathbb{R}^{L \times d_T}$ and generate responses $Y = \{y_1, y_2, \ldots, y_L\}$. We summarize widely adopted projectors into two branches:

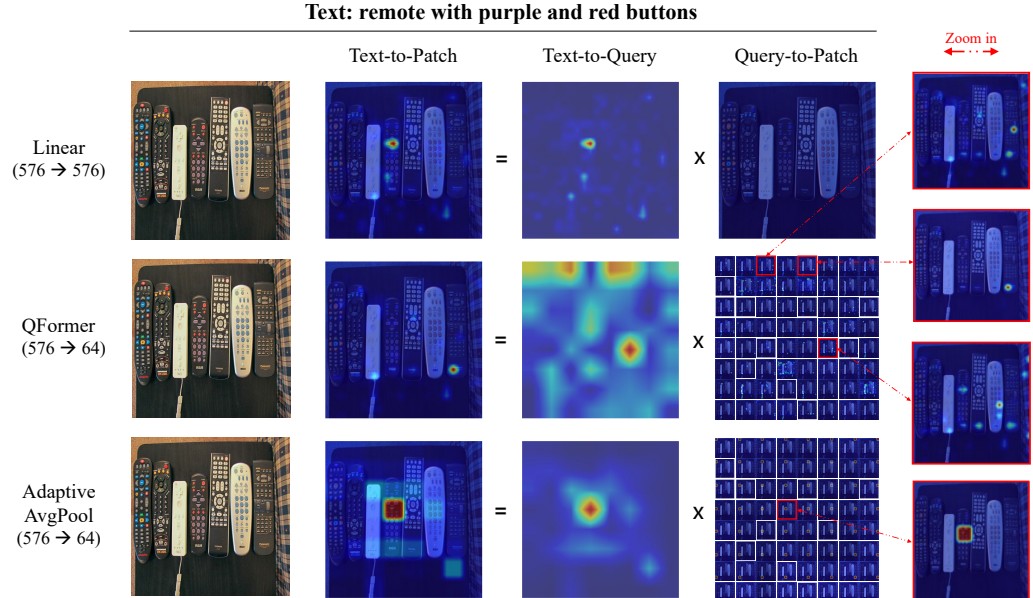

Figure 3: Visualization of the R-GAE relevance maps across the same MLLM architecture except for projector modules. The linear projector is non-compressive while the QFormer and Adaptive Average Pooling (ours) compress the original 576 vision tokens to 64 tokens. Text-to-Patch relevance reveals the effective vision semantics aligned with the LLM during image-to-text generation. For QFormer in the second row, its Query-to-Patch map discards the fine-grained visual semantics about "purple and red". This semantic deficiency is transmitted to the final Text-to-Patch map and leads to a misalignment of vision patches and textual words.

*Non-compressive Projectors* maintain the number of patch tokens $N$ and only transform the visual embedding dimension to match the dimension of the LLM, as exemplified by the linear projector (Liu et al., 2023b). The projected visual tokens can be denoted as $\mathcal{Q} \in \mathbb{R}^{N \times d_T}$.

*Compressive Projectors* reduce the number of patch tokens $N$ to a specified lesser number $M$ ($M < N$), conserving training resources. For instance, QFormer (Li et al., 2023d) learns pre-defined query tokens to compress original visual tokens. These compressed query tokens $\mathcal{Q} \in \mathbb{R}^{M \times d_T}$ are then fed into the LLM providing vision information.

For clear clarification, we distinguish the *compression* and *abstraction* concepts in this study. The *compression* refers to the reduction of vision token number in particular, whereas *abstraction* denotes the extraction of vision semantic concepts (e.g., objects and attributes, etc.).

## 2.2 R-GAE: RELEVANCE MAPS IN MLLMS DERIVED FROM GAE

We aim to employ the dissected Text-to-Query and Query-to-Patch relevance maps to examine the projector module. A straightforward attempt is utilizing the raw attention maps in MLLM layers as the relevance map (Ren et al., 2021). However, the attention map exhibits the interaction between tokens in a single layer (Chefer et al., 2021b). Instead, we require a relevance map that traces back inter-token alignment in arbitrary two layers in the MLLM, for instance, the alignment from intermediate-layer query tokens to initial-layer input patch tokens. To achieve this goal, we propose a novel R-GAE relevance map derived from the Generic Attention Explainability (GAE) (Chefer et al., 2021a). R-GAE extends the GAE method originally designed for classification tasks, to generative MLLMs, and adapts it to the typical MLLM architecture consisting of a ViT, a projector, and an LLM. The R-GAE can acquire relevance maps from any two arbitrary layers within the MLLM through propagation.

We initialize three R-GAE relevance maps including a Text-to-Patch map as $\mathbf{R}_{\mathcal{T} \to \mathcal{I}}$, a Text-to-Query map as $\mathbf{R}_{\mathcal{T} \to \mathcal{Q}}$, and a Query-to-Patch map as $\mathbf{R}_{\mathcal{Q} \to \mathcal{I}}$. Each map is an identity matrix based on the intuition that each input token's relevance score is equal in the beginning. Given an image and an instruction (e.g., "*Please describe the image with a concise sentence*"), an MLLM will generate a

textual description $Y = \{y_1, y_2, \ldots, y_L\}$ referring to the visual information. During the generation step $t$, we can cache the attention maps across the ViT, the projector and the LLM during a forward pass. Then, specifying a word class $\widehat{y}_t$ as the target prediction, we can obtain the related gradients through a backward pass. For each layer, a single R-GAE relevance map is obtained by utilizing gradients to average across the attention heads. For step $t$, we can propagate the Text-to-Query map $\mathbf{R}^t_{\mathcal{T} \to \mathcal{Q}} \in \mathbb{R}^{1 \times M}$ from the LLM's first layer to its last layer to get the final map. Similarly, the Query-to-Patch map $\mathbf{R}^t_{\mathcal{Q} \to \mathcal{I}} \in \mathbb{R}^{M \times N}$ can be propagated from the first layer to the last layer of the projector. Subsequently, the overall Text-to-Patch relevance map can be obtained by matrix multiplication of Text-to-Query and Query-to-Patch maps:

$$\mathbf{R}^t_{\mathcal{T} \to \mathcal{I}} = \mathbf{R}^t_{\mathcal{T} \to \mathcal{Q}} \times \mathbf{R}^t_{\mathcal{Q} \to \mathcal{I}} \tag{1}$$

For a complete sentence $Y$, we integrate the R-GAE relevance maps from each time step $t$ by averaging to obtain the overall visual relevance related to a factual sentence. We set the ground-truth description from an image-text pair as the target response to perform the backward process. This limits MLLMs with different projectors having the same Oracle Text-to-Patch visualization. We provide the background of GAE and the specific propagation formula of R-GAE in Appendix A. Moreover, we compare the visualization between R-GAE and original attention maps in Appendix B.

### 2.3 A Redundant Double-Abstraction Phenomenon

Based on the R-GAE maps, we analyze the different types of projectors and investigate how they affect the vision-to-language semantic alignment. For fair comparison and analysis, we train MLLMs under the same architecture, except for the projector module, and keep all other variables the same (experimental details are provided in § 3.1). We visualize the R-GAE maps of a non-compressive projector (i.e., linear layers) and a compressive projector (i.e., QFormer) in Figure 3 and draw the following findings.

***Observation 1.*** *LLMs are good visual semantic abstractors directly from patch representation.*

The non-compressive projector directly inputs the patch representation to the LLM. As shown in the first row of Figure 3, given a description containing visual objects (i.e., the remote and buttons) and attributes (i.e., purple and red), the LLM can highlight the most relevant visual regions in a fine-grained manner, as it discriminates the accurate remote with purple and red buttons among other similar remotes. This indicates that the LLM has built a strong alignment between textual and visual semantics based on the patch representation. The recent success of MLLMs (Liu et al., 2024b; Li et al., 2023b; Chen et al., 2023a) with non-compressive projection further demonstrates that the LLM itself is an efficient visual semantic abstractor. For instance, LLaVA-Next (Liu et al., 2024b), which employs a simple Multi-layer Perceptron (MLP), achieves state-of-the-art performance across diverse multimodal benchmarks.

***Observation 2.*** *Compressive projectors extract limited visual semantic concepts from patches.* Compressive projectors like QFormer pre-extract visual semantic concepts from patches and provide reduced visual tokens at the semantic level to the LLM. As the Query-to-Patch map in Figure 3 shows, the compressed 8x8 query tokens are activated with visual semantic patterns such as different remotes, buttons, control panels, and the black background board. However, the fixed number of query tokens can only cover limited visual semantic concepts from the image. Comparing the visual patterns among 64 tokens, we find that they are visually repetitive and semantically sparse. For instance, query tokens indexing $(0, 1)$ and $(2, 0)$ are nearly identical and all attend to the bottom-right panel of the right remote. These sparse query tokens lead to a deficiency in visual semantics, losing the fine-grained attribute of "purple and red buttons". Consequently, the LLM suffers from this irreversible visual semantic deficiency when re-extracting visual context in the query semantic space. As the Text-to-Query map shows, the LLM primarily attends to the query tokens indexing $(0, 2)$, $(0, 4)$, and $(4, 5)$ (framed in red), resulting in a misalignment of text words and patches verified in the Text-to-Patch map. More visualization cases are presented in Appendix D.

***Insight.*** *An inefficient MLLM system due to the double abstraction of visual semantics.*

Based on these observations, we conclude that existing compressive projectors, which learn a fixed number of query tokens, are inefficient compressors for reducing the number of vision tokens. They result in a "Double Abstraction" MLLM system, where visual semantics are first abstracted

by projectors and then re-extracted by the LLM. This dual-abstraction procedure has two main shortcomings: (i) Accumulative visual semantics loss. The projector serves as an intermediate module bridging the ViT and LLM, therefore, the visual semantics lost during the initial abstraction by the projector become a bottleneck for the MLLM system. (ii) Increased training complexity. Optimizing a projector to be an effective semantic abstractor is essential for alleviating semantic loss; however, this increases the training cost and complexity.

### 2.4 DeCo: Decoupling Vision Token Compression from Semantic Abstraction

Inspired by the analysis in §2.3, we propose a DeCo insight to **De**couple vision token **Co**mpression from semantic abstraction in MLLMs. In this approach, the compressive projectors focus on reducing the number of visual tokens with patch-level outcomes, while the LLM serves as the expert semantic abstractor. Consequently, the DeCo system only requires a simple projector that compresses visual tokens at the patch level. This design removes the intermediate semantic bottleneck and simplifies the training process.

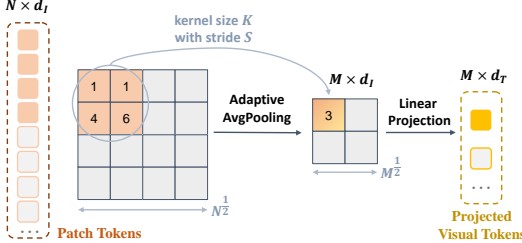

Figure 4: Visualization of the 2D Adaptive Pooling.

Based on the DeCo insight, we employ a straightforward 2D Adaptive Average Pooling (referred to as AdaptiveAvgPool) as a natural contrast to downsample the visual tokens at the patch level. As Figure 4 illustrates, given $N$ patch tokens from the ViT, the adaptive pooling can reduce the token number to a lesser square number $M$. Specifically, we reshape the N visual tokens to 2D tensors with size $(N^{\frac{1}{2}}, N^{\frac{1}{2}})$ and utilize a 2D adaptive average pooling to get compressed tokens with size $(M^{\frac{1}{2}}, M^{\frac{1}{2}})$. Subsequently, the compressed 2D tensor is flattened into $M$ tokens. These tokens are finally projected by the linear layer to match the textual embedding dimension, serving as visual inputs to the LLM. During compression, the adaptive pooling [1] automatically calculates the stride $S$ and kernel size $K$ in a parameter-free mode. It averages patches in a spatial $K \times K$ window into a mixed token. In essence, the 2D AdaptiveAvgPool merges the spatial neighbor patch tokens which tend to have high visual redundancy.

As illustrated in the third row of Figure 3, the Query-to-Patch mapping of the AdaptiveAvgPool projector forms a 2D grating pattern. It uniformly down-samples the grouped patches over the 2D spatial space of the original image. This uniform patch-level sampling preserves dense visual context compared to the QFormer abstractor. For instance, the compressed token indexed at $(3, 3)$, highlighted in the red frame, retains the fine-grained representation of the "purple and red buttons". Subsequently, the LLM can attend to the accurate visual region by leveraging the visual context from the AdaptiveAvgPool, as shown in the Text-to-Patch map. Furthermore, the Text-to-Patch maps of the linear projector and AdaptiveAvgPool are nearly identical. This similarity reveals that the AdaptiveAvgPool projector achieves a superior combination of (i) effectiveness, approximating the linear projector in preserving visual context, and (ii) efficiency, reducing the number of vision tokens, similar to the QFormer abstractor.

## 3 Quantitative Results

In this section, we qualitatively validate the simple AdaptiveAvgPool following the DeCo insight, by comparing it with prevailing compressive projectors, including QFormer, C-Abstractor, and D-Abstractor (Cha et al., 2024), in terms of both effectiveness and efficiency.

### 3.1 Experiment Setting

**Training data and Evaluation.** We utilize the open-sourced 558K pre-training data (sourced from LAION (Schuhmann et al., 2021), Conceptual Captions (Changpinyo et al., 2021) and SBU Captions (Ordonez et al., 2011)) and 665K instruction-following data (containing LLaVa Synthetic

---

[1]Apply the torch.nn.AdaptiveAvgPool2d function in the PyTorch framework.

Table 1: Overall performance compared to existing compressive projectors including Linear Projector (Liu et al., 2023a), QFormer (Li et al., 2023d), and C-Abstractor/D-Abstractor (Cha et al., 2024). All results are conducted under the same architecture and settings. We All compressive projectors reduce the vision token number (#V) from 576 to 144. * indicates reproduced results using LoRA while † denotes the full-training results reported in LLaVA v1.5. Avg$^N$ means an average of normalized benchmark scores. The best and second-best results are **bolded** and underlined, respectively.

| Projectors | #V | SEED$^I$ | MME$^P$ | POPE | Refcoco | Refcoco+ | Refcocog | VizWiz | VQA$^{v2}$ | GQA | VQA$^{Text}$ | Avg$^N$ |
|---|---|---|---|---|---|---|---|---|---|---|---|---|
| Linear$^†$ | 576 | 66.2 | 1524.6 | 86.4 | 54.4 | 47.8 | 49.8 | 53.6 | 76.3 | 60.0 | 58.9 | 63.0 |
| Linear$^*$ | 576 | 65.1 | 1338.6 | 86.8 | 46.9 | 41.6 | 46.3 | 50.2 | 74.9 | 56.5 | 58.4 | 59.4 |
| QFormer | 144 | 55.3 | 1312.7 | 79.0 | 15.1 | 10.5 | 11.6 | **51.2** | 65.6 | 48.6 | 50.7 | 45.3 |
| C-Abstractor | 144 | 60.5 | **1411.8** | 84.5 | 40.6 | 34.3 | 38.4 | 47.8 | 70.9 | 52.6 | 55.9 | 55.6 |
| D-Abstractor | 144 | 60.0 | 1313.2 | 84.6 | 32.9 | 27.6 | 32.4 | 49.7 | 71.1 | 53.1 | 55.1 | 53.2 |
| DeCo (Ours) | 144 | **62.8** | 1373.4 | **85.9** | **43.4** | **38.5** | **39.3** | 49.7 | 74.0 | **54.1** | 56.2 | **57.3** |

Table 2: Performance comparison on more fine-grained benchmarks includes the informative diagram benchmark AI2D (Chen et al., 2024) and ChartQA (Masry et al., 2022) for chart-related question answering, document understanding benchmarks such as DocVQA (Mathew et al., 2021), and science topic question answering with ScienceQA (Lu et al., 2022).

| Projectors | AI2D | ChartQA | DocVQA | SciQA$^{img}$ | Avg$^N$ |
|---|---|---|---|---|---|
| QFormer | 52.4 | 12.4 | 15.8 | 68.4 | 37.3 |
| C-Abstracter | 53.9 | 14.3 | 19.0 | 69.1 | 39.1 |
| D-Abstractor | 52.2 | 14.2 | 19.4 | 68.0 | 38.5 |
| DeCo (Ours) | 52.8 | 15.4 | 20.9 | 68.4 | **39.4** |

Table 3: Vision spatial understanding capability: Position (POS) for MME, Spatial Relationship (SR), Object Localization (OL), and Physical Relation (PR) for MM-Bench, and Spatial Relation (SR) and Instance Location (IL) for SEED-Bench.

| Projector | #V | MME | MMB | | | SEED | | Avg |
|---|---|---|---|---|---|---|---|---|
| | | POS | SR | OL | PR | SR | IL | |
| Linear | 576 | 123.3 | 20.0 | 51.9 | 33.3 | 50.2 | 59.6 | 56.4 |
| QFormer | 144 | 73.3 | 17.8 | 33.3 | 33.3 | 39.0 | 48.9 | 40.9 |
| C-Abstractor | 144 | 116.7 | 15.6 | 42.0 | **54.2** | 43.5 | 54.4 | 54.4 |
| DeCo (Ours) | 144 | **116.7** | **24.4** | **48.1** | 41.7 | **46.6** | **58.5** | **56.0** |

Data (Liu et al., 2023b), VQA$^{v2}$ (Goyal et al., 2017), GQA (Hudson & Manning, 2019), OK-VQA (Marino et al., 2019), OCR-VQA (Mishra et al., 2019), A-OKVQA (Schwenk et al., 2022), TextCaps (Sidorov et al., 2020), RefCOCO (Yu et al., 2016), Visual Genome (Krishna et al., 2017) and ShareGPT (ShareGPT, 2023)) following LLaVA v1.5 (Liu et al., 2023a). For evaluation, we measure model performance spanning three aspects. *Multimodal LLM Benchmarks* including SEED-Bench (Li et al., 2023c) (report image-only set as SEED$^I$), MME (Fu et al., 2023) (report perception set as MME$^P$) and POPE (Li et al., 2023g) are specially designed for instruction-following MLLMs. *Visual Localization* task encompassing RefCOCO, RefCOCO+, and RefCOCOg (Kazemzadeh et al., 2014; Yu et al., 2016) is to measure the bounding box prediction accuracy. *Open-Ended Visual Question Answering* task consisting of VizWiz (Bigham et al., 2010), VQA$^{v2}$ (Goyal et al., 2017), GQA (Hudson & Manning, 2019) and TextVQA (Singh et al., 2019) aims to evaluate visual reasoning capability.

**Implementation Details.** DeCo is primarily built on the LLaVA v1.5 framework, encompassing model architectures, training data, and training strategies. We replace the original two-layer MLP projector with QFormer (Li et al., 2023d), C-Abstractor (Cha et al., 2024), D-Abstractor (Cha et al., 2024) and AdaptiveAvgPool respectively for fair comparison. The default configuration includes a CLIP ViT-L/14 336px (Radford et al., 2021) and Vicuna v1.5 7B (Chiang et al., 2023) with a two-stage training strategy. The first pre-training stage updates only the projector while the second instruction-tuning stage optimizes both the projector and the LLM using LoRA (Hu et al., 2022). The main results are derived from this default configuration. Additionally, we conduct generalization experiments using a more lightweight setup that involves only the instruction tuning stage as outlined in PRISM (Karamcheti et al., 2024). Specific training hyper-parameters are detailed in Appendix C.

## 3.2 COMPARED WITH EXISTING PROJECTORS

To showcase the efficiency and effectiveness of the DeCo method with AdaptiveAvgPool, we compare it with common projectors including the Linear projector (Liu et al., 2023b), QFormer (Li et al., 2023d), C-Abstractor (Cha et al., 2024), and D-Abstractor(Cha et al., 2024).

**Performance Effectiveness.** Table 1 presents the overall performance of different projectors. The non-compressive linear projector preserves all vision information and achieves the best overall performance. In the compressive projector category, DeCo outperforms existing solutions across

Table 4: Comprehensive comparison between the C-Abstractor (C-Abstr) and Adaptive Averaging Pooling (AvgPool) across various settings including different vision backbones, image resolutions and LLMs. All experiments are conducted on the one-stage instruction tuning (665K data) referring to PRISM (Karamcheti et al., 2024) to speed up training. *Res.* denotes image resolution. *Compress.* means the compression ratio of each projector from the raw visual token number to the projected vision token number.

| | ViT | LLM | Res. | Compress. | Project. | POPE | Refcoco / + / g | VizWiz | VQA$^{v2}$ | GQA | VQA$^{Text}$ |
|---|---|---|---|---|---|---|---|---|---|---|---|
| B1 | SigLIP ViT-SO | Phi-2 (2.7B) | 224 | 256->144 | C-Abstr | 66.1 | 11.5 / 6.5 / 8.4 | 18.7 | 47.8 | 42.0 | 34.6 |
| | | | | | AvgPool | **84.1** | **21.5 / 13.6 / 15.6** | **34.5** | **68.1** | **52.6** | **41.2** |
| B2 | CLIP ViT-L | Phi-2 (2.7B) | 336 | 576->144 | C-Abstr | 73.7 | 11.8 / 7.3 / 6.9 | 18.0 | 52.5 | 45.3 | 36.7 |
| | | | | | AvgPool | **84.5** | **15.0 / 9.3 / 8.8** | **28.4** | **64.6** | **48.9** | **40.8** |
| B3 | SigLIP ViT-SO | Phi-2 (2.7B) | 384 | 729->144 | C-Abstr | 78.8 | 12.9 / 8.2 / 7.7 | **41.3** | 53.2 | 45.1 | 35.4 |
| | | | | | AvgPool | **81.7** | **17.4 / 11.4 / 11.0** | 39.5 | **60.3** | **48.0** | **40.2** |
| B4 | DINOv2+SigLIP | Phi-2 (2.7B) | 384 | 729->144 | C-Abstr | 52.6 | 13.5 / 6.6 / 7.5 | **29.2** | 40.9 | 36.3 | 34.9 |
| | | | | | AvgPool | **85.7** | **24.9 / 17.3 / 21.6** | 24.0 | **63.9** | **52.6** | **39.2** |
| B5 | DINOv2+SigLIP | Qwen-Chat (0.5B) | 384 | 729->144 | C-Abstr | **49.9** | 8.7 / 4.3 / 7.6 | 17.7 | 53.8 | 45.1 | 28.9 |
| | | | | | AvgPool | **49.9** | **12.9 / 9.7 / 11.2** | **25.3** | **58.3** | **46.5** | **31.4** |
| B6 | DINOv2+SigLIP | Vicuna-v1.5 (7B) | 384 | 729->144 | C-Abstr | 86.0 | 31.7 / 25.5 / 29.2 | 39.1 | 62.6 | 52.3 | 46.5 |
| | | | | | AvgPool | **87.0** | **42.3 / 33.1 / 37.6** | **52.2** | **69.8** | **55.4** | **49.3** |

most benchmarks. Specifically, DeCo achieves gain margins of SEED$^I$ +2.3 and POPE +1.3 in the instruction-following MLLM benchmarks, RefCOCO/RefCOCO+/RefCOCOg +2.8/4.2/0.9 for visual localization, and VQA$^{v2}$ +3.9, GQA +1.0, VQA$^{Text}$ +0.3 for open-ended visual question answering. The superior results of DeCo under the same compression ratio (576->144) demonstrate that naive compression at the patch level effectively transmits visual context while reducing the token number. Among the existing projectors, the locality-enhanced C-Abstractor produces results comparable to DeCo. Additionally, we observe that QFormer performs poorly on the visual localization task, particularly in predicting visual coordinates. This poor performance is due to the loss of spatial locality during projector compression, resulting in cumulative spatial context deficiency. Besides, as Table 2 shows, AdaptiveAvgPool also performs the best on fine-grained tasks. The remarkable result gain on the ChartQA (+7.69%) and DocVQA (+7.73%) dataset requiring fine-grained visual cues (e.g., flowchart labels, plot axes) reveals that AdaptiveAvgPool is efficient in both widely-used general benchmarks and more challenging sets.

**Training Efficiency.** Besides the remarkable performance, DeCo also has efficiency advantages because it conducts parameter-free compression clarified in § 2.4. Among existing compressive projectors, the sub-optimal C-Abstractor comprises 3-layer ResNet blocks (Xie et al., 2017), the adaptive average pooling and another 3-layer ResNet blocks. Meanwhile, we adopt a two-layer QFormer consisting of a self-attention and a cross-attention layer initialized from the BLIP-2 (Li et al., 2023d) pretraining weights. Compared with them, the AdaptiveAvgPool in DeCo method is more lightweight and efficient. Figure 5 depicts that DeCo has faster training convergence during pre-training.

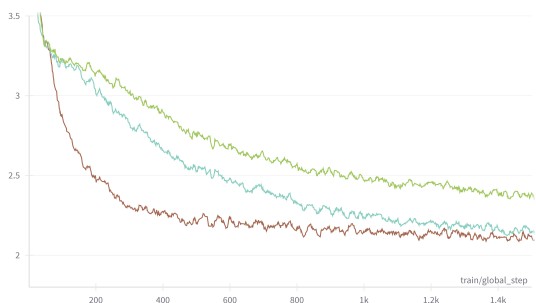

Figure 5: Pre-training loss convergence of AdaptiveAvgPool (brown), C-Abstractor (blue) and QFormer (green).

**Spatial Locality Reservation.** Spatial understanding capability in vision modality is essential to achieve accurate visual location, fine-grained vision reasoning, object relation perception and etc. We verify the spatial understanding capability of DeCo in Table 3 across six spatial understanding tasks from MLLM benchmarks. As Honeybee (Cha et al., 2024) points out, the vanilla resampler architecture like QFormer will lose the visual spatial locality, therefore, it obtains a low average score of 40.9. The locality-enhanced projector, i.e., C-Abstractor, has remarkable improvements and achieves 54.4. Overall, the DeCo with AdaptiveAvgPool well reserves the significant spatial context

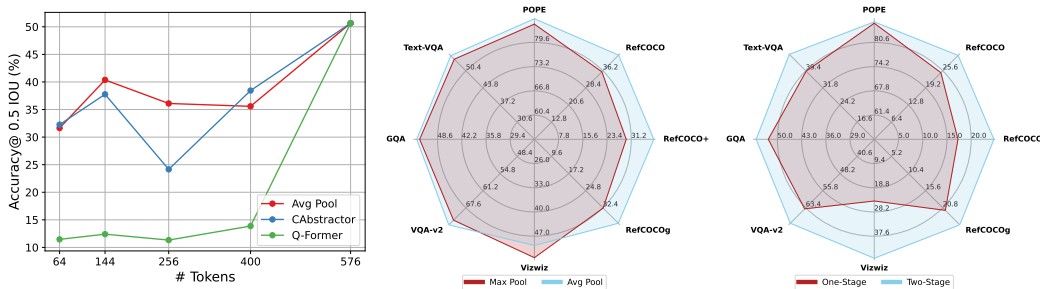

Figure 6: Compression ratio reducing 576 tokens to 400/256/144/64 tokens.

Figure 7: Comparison of Max Pooling and Average Pooling.

Figure 8: Comparison of one-stage and two-stage training.

and achieves the closest score (56.0) to the linear projector (56.4). This benefits from the kernel and stride operation of 2D AdaptiveAvgPool similar to the convolutional network (Li et al., 2021).

### 3.3 GENERALIZATION RESULTS

To explore the performance of DeCo under different configurations, we select varied vision backbones, image resolutions and LLMs, and report results in Table 4. To speed up training, all results are obtained through the one-stage training (i.e., instruction tuning) according to PRISM (Karamcheti et al., 2024). We select the most comparative baseline C-Abstractor (refer to Table 1) as a reference.

For vision backbones (B2, B3, and B4), we adopt the CLIP ViT-L, SigLIP ViT-SO (Zhai et al., 2023), and the DINOv2 (Oquab et al., 2023)+SigLIP ensemble in embedding dimension. For scaling image resolution (B1 and B3), we compare 224px and 384px image inputs using the SigLIP ViT-SO backbone. For LLMs (B4, B5, and B6), we employ three levels of model scope, including Qwen-Chat-0.5B (Bai et al., 2023a), Phi-2-2.7B (Javaheripi et al., 2023), and Vicuna-v1.5 (Chiang et al., 2023).

The overall results in Table 4 under six different settings demonstrate the robustness of DeCo as a compressive projector across diverse MLLM architectures. It surpasses the C-Abstractor notably in almost all metrics and all settings.

### 3.4 ABLATION STUDY

**Compression Ratio Analysis.** There is a trade-off between visual information deficiency and training cost based on the compression ratio. In Figure 6, we compress the visual tokens from $24 \times 24$ to $20 \times 20$, $16 \times 16$, $12 \times 12$, and $8 \times 8$ respectively, and report the average Accuracy@IoU=0.5 on the visual localization task. Results reveal that a quarter compression from $24 \times 24$ to $12 \times 12$ provides the best balance for AdaptiveAvgPool.

**Average Pooling vs. Max Pooling.** Average pooling and max pooling are two widely-used downsampling operations. We compare these two operations in the DeCo method in Figure 7. Results show that adaptive average pooling performs better across almost all metrics, especially visual localization. The reason is that the averaging operation integrates each patch within the kernel-size window and can serve more visual context.

**One-Stage vs. Multi-Stage Training.** PRISM (Karamcheti et al., 2024) indicates simple linear projectors only require one-stage instruction tuning. Inspired by this, we compare the one-stage and two-stage training results of DeCo and find that two-stage training is recommended, as shown in Figure 8.

## 4 RELATED WORK

**Multimodal Large Language Models.** The development of large vision-language models has accelerated recently (OpenAI, 2023; Reka, 2024; Gemini Team, 2023; Li et al., 2023e). Flamingo (Alayrac et al., 2022; Awadalla et al., 2023) and IDEFICS (Laurençon et al., 2023) have showcased the effectiveness of consolidating LLMs with vision encoders. The Q-Former from BLIP-2 (Li et al., 2023d) has helped bridge the gap between the visual and text modalities. InstructBLIP (Dai et al.,

2023), Ying-VLM (Li et al., 2023f) and MM-ICL (Zhao et al., 2023) further integrate instructions into the vision-text alignment process for improved in-context learning ability (Dong et al., 2022). Various approaches have been proposed to align visual encoders and LLMs effectively. MiniGPT-4 (Zhu et al., 2023) and LLaVA (Liu et al., 2023b;a) use a single projection layer, while mPLUG-Owl (Ye et al., 2023) adopts LoRA tuning (Hu et al., 2022; Ma et al., 2024), showing promising results. Qwen-VL-Chat (Bai et al., 2023b) has scaled up multi-modal pre-training with more datasets. Fuyu-8 (Bavishi et al., 2023) proposes a new architecture by segmenting images into pixel patches, treating them as visual tokens to train a conditional multi-modal language model directly. However, these works employ projector modules empirically or simply refer to the final performance of the MLLMs on downstream tasks without conducting an in-depth analysis of the projectors' effectiveness. In this paper, we examine this significant component by tracking the vision-and-language semantic flow within MLLMs. We visualize the internal patterns learned by projectors and highlight their drawbacks, offering valuable insights for future development.

**Transformer Explainability.** Explainability tools have been widely explored for Transformers to better visualize their inner decision-making processes. Raw attention maps in Transformers usually provide interpretations for a single layer. Abnar et al. (Abnar & Zuidema, 2020) combine the attention scores across multiple layers and propose the rollout method. Chefer et al. (Chefer et al., 2021b) introduce the relevance map through information propagation from all layers and components in Transformers. LRP (Voita et al., 2019) captures the relative importance between different attention heads using gradients. Casual Interpretation (Rohekar et al., 2023) can identify the most important input tokens corresponding to the model output. However, these methods are only applicable to Transformers with self-attention layers. As an alternative, the GAE (Chefer et al., 2021a) method extends the propagated relevance maps to bi-modal scenarios with cross-attention layers. Moreover, several studies (Aflalo et al., 2022; Liu et al., 2023c; Lyu et al., 2022; Ramesh & Koh, 2022; Swamy et al., 2024) focus on the multimodal system interpretation. Recently, LVLM-Interpret (Ben Melech Stan et al., 2024) has developed an interactive application for interpreting MLLMs. Despite these efforts, in-depth explainability of existing MLLMs is rarely explored. In this study, we propose the R-GAE method derived from GAE for MLLMs to investigate how projector modules affect the vision-and-language semantic alignment of MLLMs.

## 5 CONCLUSION

We introduce DeCo to decouple visual token compression from semantic abstraction. It is motivated by the "Double Abstraction" problem of existing projectors disentangling the Text-to-Patch, Text-to-Query and Query-to-Patch R-GAE maps in the vision-and-language semantic alignment. The DeCo method simplifies existing compressive projectors with a naive AdaptiveAvgPool, which downsamples spatial vision tokens directly at the spatial level. Experiments across diverse configurations demonstrate the efficiency, effectiveness, and robustness of DeCo. Eventually, the intuition of "DeCo" is not limited to the specific AdaptiveAvgPool projector design, there is great potential to improve it to perform more effectively under more demanding scenarios like high compression ratio.

## LIMITATIONS

We present limitations in this work to facilitate future research. Firstly, the AdaptiveAvgPool adopted in the DeCo method may cause severe visual information deficiency in an increasingly high compression ratio compared to semantic-level compression projectors. In a high-compression scenario, the averaging pooling will erase the fine-grained visual context in a kernel scope. Secondly, the superiority of DeCo lies in a limited training resource application including limited GPUs to train a long visual token sequence and limited training data to optimize a desirable semantic QFormer-type projector. Otherwise, when have abundant training resources, the architecture of projectors tend to be insignificant in an MLLM system as pointed out in the MM1 (McKinzie et al., 2024).

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

# A DETAILS OF R-GAE EXPLAINABILITY TOOL

## A.1 BACKGROUND: GAE EXPLAINABILITY FOR TRANSFORMER LAYERS

The Generic Attention Explainability (GAE) (Chefer et al., 2021a) is a powerful method to interpret predictions for bi-modality Transformer-based architectures. It has the advantage of acquiring the relevance map from two arbitrary layers in the Transformer through propagation. Essentially, the GAE method generates a relevance map $\bar{\mathbf{A}}$ for each self-attention layer or cross-attention layer by integrating raw attention maps and gradients. Then it aggregating the relevance maps of all layers into a overall single map $\mathbf{R}$. Formally, denote a Transformer architecture as $\phi$, its attention map of each layer as $\mathbf{A}$, the input modality tokens as $I \in \mathbb{R}^{N \times d}$ and the output predict class as $y$. We aim to visualize the relevance map $\mathbf{R}_{y \to I} \in \mathbb{R}^N$ from class $y$ to input tokens $I$. Take the self-attention layer as an example, the relevance map $\bar{\mathbf{A}}$ for each layer and the propagation of final map $\mathbf{R}_{y \to I}$ are termed as:

$$\bar{\mathbf{A}} = \mathbb{E}_h((\nabla \mathbf{A} \odot \mathbf{A})^+),\tag{2}$$

$$\mathbf{R} = \mathbf{R} + \bar{\mathbf{A}} \cdot \mathbf{R},\tag{3}$$

where each layer's attention map $\mathbf{A}$ can be obtained through a forward pass and the related gradient $\nabla \mathbf{A} := \frac{\partial \phi(y)}{\partial \mathbf{A}}$ can be cached during a backward pass. $\odot$ is the Hadamard product, $(\cdot)^+$ represents the operation of setting negative values to 0, and $\mathbb{E}_h$ is the mean across the attention heads dimension. The overall map $\mathbf{R}$ is initialized as the identity matrix with the intuition that each input token's relevance score is identical in the beginning. The propagation Formula 3 updates the $\mathbf{R}$ from a start layer $L_s$ to an end layer $L_e$ ($L_e > L_s$) in the Transformer. The cross-attention propagation is similar, which maintains two relevance matrices for two modalities and updates them through the layer interaction. Please refer to the details of the propagation formula across cross-attention layer from the original paper.

A.2   R-GAE Propagation for MLLMs

The traditional GAE map is designed for a classification task with the special $CLS$ token. We adapt it to MLLM architectures and propose the R-GAE explainability tool. As Figure 2 shows, a typical MLLM architecture comprises a Vision Transformer (ViT) $\phi_v$ to acquire patch-level visual representations $\mathcal{I} \in \mathbb{R}^{N \times d_I}$ (containing $N$ patches), a projector $\phi_p$ to transform visual representations into the textual embedding space as $\mathcal{Q}$, and an LLM $\phi_t$ that handles both vision and instruction tokens to output hidden states $\mathcal{T} \in \mathbb{R}^{L \times d_T}$ and generate responses $Y = \{y_1, y_2, \ldots, y_L\}$. We summarize widely adopted projectors into two branches:

*Non-compressive Projectors* maintain the number of patch tokens $N$ and only transform the visual embedding dimension to match the dimension of the LLM, as exemplified by the linear projector (Liu et al., 2023b). The projected visual tokens can be denoted as $\mathcal{Q} \in \mathbb{R}^{N \times d_T}$.

*Compressive Projectors* reduce the number of patch tokens $N$ to a specified lesser number $M$ ($M < N$), conserving training resources. For instance, QFormer (Li et al., 2023d) learns pre-defined query tokens to compress original visual tokens. These compressed query tokens $\mathcal{Q} \in \mathbb{R}^{M \times d_T}$ are then fed into the LLM providing vision information.

We initialize three GAE relevance maps including a Text-to-Patch map as $\mathbf{R}_{\mathcal{T} \to \mathcal{I}}$, a Text-to-Query map as $\mathbf{R}_{\mathcal{T} \to \mathcal{Q}}$, and a Query-to-Patch map as $\mathbf{R}_{\mathcal{Q} \to \mathcal{I}}$. As Figure 3 depicts, given an image and an instruction (e.g., "*Please describe the image with a concise sentense*"), an MLLM will generate a textual description $Y = \{y_1, y_2, \ldots, y_L\}$ referring to the visual information. During the generation step $t$, we can cache the attention map $\mathbf{A}_v, \mathbf{A}_p, \mathbf{A}_t$ across the ViT, the projector and the LLM during a forward pass. Then specifying a word class $\widehat{y}_t$ as the target predict, we can get the gradients $\nabla \mathbf{A}_t$, $\nabla \mathbf{A}_p, \nabla \mathbf{A}_v$ in each module through a backward pass. The LLM module in MLLMs substantially contains self-attention layers, therefore, we can propagate the $\mathbf{R}_{\mathcal{T} \to \mathcal{Q}}^t \in \mathbb{R}^{1 \times M}$ according to Formula 3 from LLM's first layer to its last layer. The QFormer-type projector consisting of self-attention and cross-attention layers can also be propagated similarly to get $\mathbf{R}_{\mathcal{Q} \to \mathcal{I}}^t \in \mathbb{R}^{M \times N}$. Subsequently, the overall text-to-patch relevance map can be obtained by matrix multiplication of text-to-query and query-to-patch maps:

$$\mathbf{R}_{\mathcal{T} \to \mathcal{I}}^t = \mathbf{R}_{\mathcal{T} \to \mathcal{Q}}^t \times \mathbf{R}_{\mathcal{Q} \to \mathcal{I}}^t \tag{4}$$

For a complete sentence $Y$, we integrate the GAE relevance maps from each time step $t$ by averaging them to obtain the overall visual relevance related to a factual sentence. The final three maps are formulated as followings, in which $\mathbf{R}_{\mathcal{T} \to \mathcal{I}} \in \mathbb{R}^{1 \times N}$, $\mathbf{R}_{\mathcal{T} \to \mathcal{Q}} \in \mathbb{R}^{1 \times M}$, and $\mathbf{R}_{\mathcal{Q} \to \mathcal{I}} \in \mathbb{R}^{M \times N}$.

$$\mathbf{R}_{\mathcal{T} \to \mathcal{Q}} = \frac{1}{L} \sum_{t=1}^{L} \mathbf{R}_{\mathcal{T} \to \mathcal{Q}}^t, \qquad \mathbf{R}_{\mathcal{Q} \to \mathcal{I}} = \frac{1}{L} \sum_{t=1}^{L} \mathbf{R}_{\mathcal{Q} \to \mathcal{T}}^t, \qquad \mathbf{R}_{\mathcal{T} \to \mathcal{I}} = \frac{1}{L} \sum_{t=1}^{L} \mathbf{R}_{\mathcal{T} \to \mathcal{I}}^t \tag{5}$$

For non-compression projectors maintaining the number of original patches, such as linear layers, the Query-to-Patch map is an identity mapping based on the one-to-one correspondence between queries and patches. Consequently, the Query-to-Image map visualizes the original image consisting of 576 patches. The Text-to-Query map is obtained in the same manner as in the QFormer, which propagates from the R-GAE maps in the Language Model (LLM).

For the AdaptiveAvgPool projector in the DeCo method, a 2D spatial down-sampling mapping is constructed from the original tokens to the compressed tokens. For an operation window with kernel size $K$, the merged token is assigned a relevance score equal to $1/K^2$ of the sum of the relevance scores of each raw token within the window. The corresponding Query-to-Patch map can be calculated using this simple mapping rule. Similar to the QFormer, the Text-to-Query map is obtained from the LLM layers.

# B   Comparison between R-GAE and Raw Attention Maps

The R-GAE map offers two advantages over raw attention maps: (i) it demonstrates better explainability (Chefer et al., 2021a) by integrating both attention maps and gradients, and (ii) it can track token relevance from a target layer (e.g., output textual tokens) to the first layer (e.g., original patch tokens). In contrast, the attention map commonly used from the last layer of the Large Language

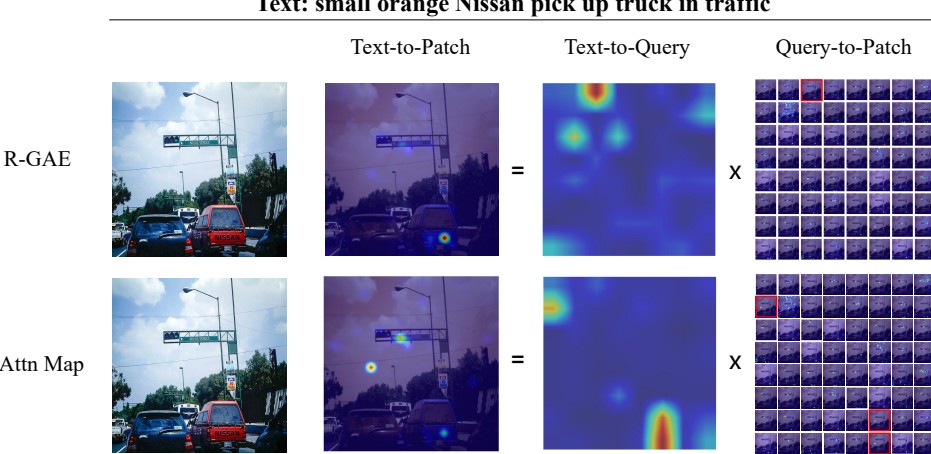

Figure 9: Comparison between the R-GAE and raw attention map explainability on the same case from the QFormer projector, which compresses 576 vision tokens to 64 query tokens.

Model (LLM) can only show the relevance of mixed tokens in that layer, where the tokens related to the vision input position or the output word position have incorporated the semantics of other tokens through attention operation in previous layers.

Figure 9 visualizes the R-GAE relevance maps and the raw attention maps for a comparative analysis. The Query-to-Patch map of raw attention is obtained from the last cross-attention layer in the QFormer, while the Text-to-Query map of raw attention is derived from the last layer in the LLM. By visualizing the same model and image-text pair, it becomes evident that R-GAE provides a more interpretable representation of the inner vision-language alignment of an MLLM. In contrast, the raw attention map highlights an unrelated visual patch, such as the sky, which introduces an additional error in the explainability procedure when analyzing semantic alignment. An in-depth analysis reveals that the error in the raw attention map primarily originates from the Text-to-Query map of the last LLM layer. This can be attributed to the fact that the LLM consists of 32 self-attention layers, and the relevance among query tokens and text tokens in the last layer has deviated due to the fusion of semantics from other tokens in previous layers. On the other hand, the Query-to-Patch map exhibits relatively similar characteristics to the R-GAE map. This similarity can be explained by the architecture of the QFormer, which only employs a single cross-attention layer, thus minimizing the influence of token fusion across layers for raw attention.

## C   TRAINING HYPER-PARAMETERS

**Architecture of Used Projectors.**

1. C-Abstractor comprises 3-layer ResNet blocks (Xie et al., 2017), the adaptive average pooling and another 3-layer ResNet blocks.

2. D-Abstractor leverages Deformable Attention (Zhu et al., 2020) to replace the vanilla attention and conduct well-designed initialization of query tokens. We adopt a two-layer D-Abstractor.

3. QFormer is a two-layer BERT (Devlin et al., 2019) architecture same as the the BLIP-2 (Li et al., 2023d) and we load the BLIP-2 pre-training weights as an initialization.

4. Linear projector is a two-layer MLP with the GELU activation same as the LLaVA v1.5 (Liu et al., 2023a).

5. AdaptiveAvgPool is parameter-free, we utilize a two-layer MLP as the linear projector to map the vision feature dimension to the LLM's.

**Training Parameters.** Our experiments are conducted under two primary training settings. The main experiments are built on the LLaVA v1.5 framework, as shown in Table 5. The generalization experiments are constructed using a more lightweight setup that involves only the instruction tuning stage, referring to the PRISM (Karamcheti et al., 2024) approach. Specific training hyperparameters are detailed in Table 6.

Table 5: Hyper-parameters of main experiments.

| Hyperparameter | Pretrain | Finetune |
|---|---|---|
| batch size | 256 | 128 |
| lr | 1e-3 | 2e-5 |
| lr schedule | cosine decay | |
| lr warmup ratio | 0.03 | |
| weight decay | 0 | |
| epoch | 1 | |
| optimizer | AdamW | |
| DeepSpeed stage | 2 | 3 |

Table 6: Hyper-parameters of generalization experiments.

| Hyperparameter | Value |
|---|---|
| Batch Size | 128 |
| Max Gradient Norm | 1.0 |
| Weight Decay | 0.1 |
| Learning Rate | 2e-5 |
| Optimizer | AdamW |
| Scheduler | Warmup & Cosine Decay |
| Warmup Ratio | 0.03 |

## D    MORE R-GAE RELEVANCE MAPS

Figure 10 presents additional visualized cases of the R-GAE relevance map across different projectors.

## E    BROADER IMPACTS

Our work utilizes off-the-shelf frozen LLMs, which means it shares some of their intrinsic drawbacks, such as generating hallucinated, ungrounded text or biased outputs. We mitigate these issues by enhancing the model's grounding in both visual and instruction inputs. Additionally, our training dataset includes 40K examples of safety data sourced from ShareGPT, instructing the models to refuse responses to toxic, inappropriate, or otherwise unsafe inputs. However, we do not recommend applying our models to any downstream applications without a prior assessment of safety and fairness specific to that application.

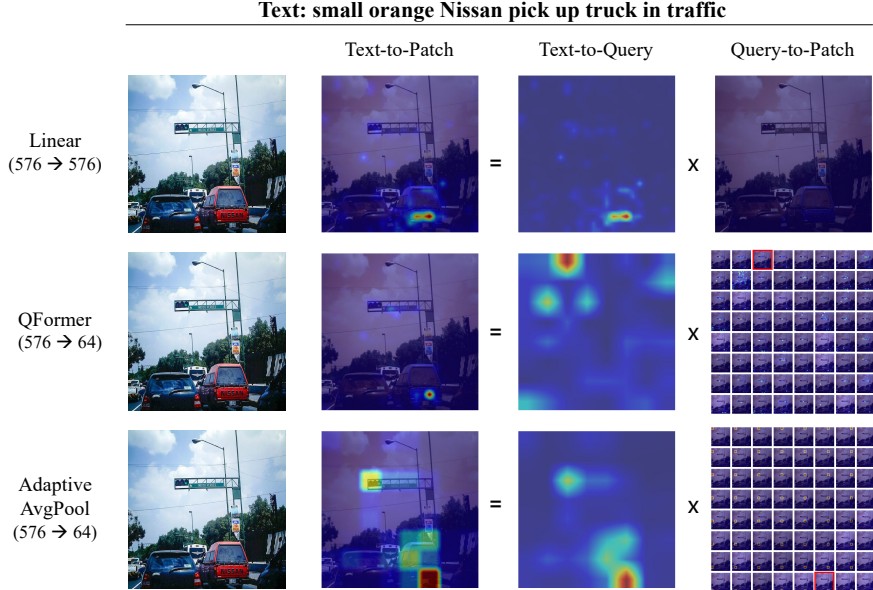

(a) R-GAE maps related to the generated text "small orange Nissan pick up truck in traffic". In this case, all projectors reserve the effective visual representation and translate it to the LLM. Specifically, the Qformer-based MLLM attends to the query indexing $(0, 2)$ which highlights the "Nissan" semantics on the image. This indicates extracting effective visual semantic concepts in the first abstraction by the QFormer is important for the traditional compressive projectors.

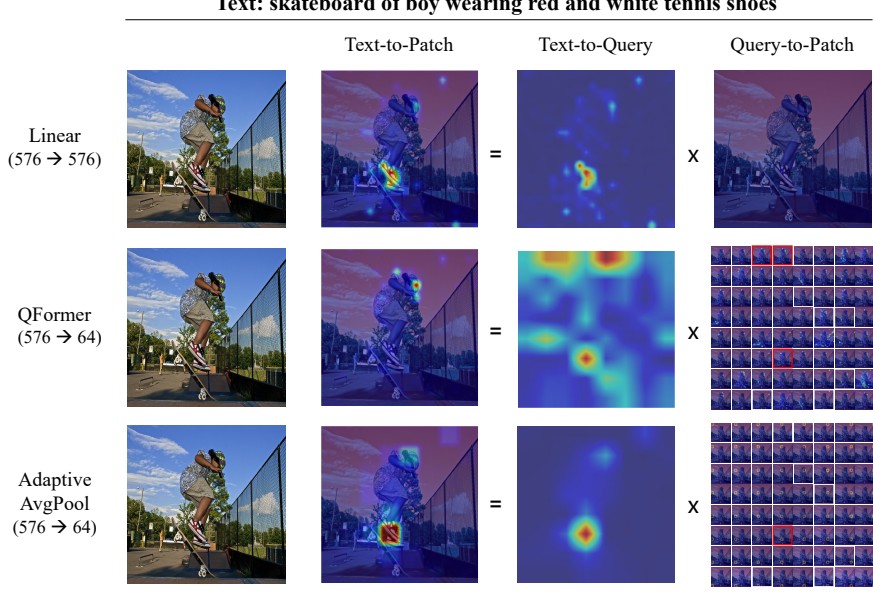

(b) R-GAE maps related to the generated text "skateboard of boy wearing red and white tennis shoes" are shown in Figure 10. In this case, the QFormer-based MLLM fails to attend to the relevant patches with the "red and white tennis shoes" attributes. In contrast, both the linear projector and the AdaptiveAvgPool highlight the correct patches.

Figure 10: Visualization of additional R-GAE relevance maps. The linear projector is non-compressive, while the QFormer and Adaptive Average Pooling (AdaptiveAvgPool) compress the original 576 vision tokens to 64. For better visualization, the highlighted query tokens from the text are framed in red.

