# OpenReview forum: "DeCo: Decoupling Token Compression from Semantic Abstraction in Multimodal Large Language Models"
_ICLR.cc/2025/Conference — ICLR 2025 Conference Withdrawn Submission_

### Official Review · Reviewer_YoWL · 2024-10-16

**Soundness:** 1
**Presentation:** 1
**Contribution:** 1
**Rating:** 1
**Confidence:** 5

**Summary:**

This paper proposes a novel visual projector, named DeCo. It adopts a 2D Adaptive Pooling to reduce the number of visual tokens. Besides, This article designs a novel R-GAE explainability tool to deeply analyze double abstraction in MLLM. The experimental results showed that DeCo achieves superior results across diverse benchmarks.

**Strengths:**

1. DeCo deeply analyzes the problem of double abstraction in MLLM, providing good insights for the MLLM community. Moreover, DeCo reveals why QFormer fails to achieve high performance.

2. R-GAE is able to facilitate the analysis of MLLM, which helps the researchers to analyze the problems that existed in MLLM.

**Weaknesses:**

1. The framework and methodology lack innovation. 2D Adaptive Average pooling is not a novel approach to reduce the number of visual tokens. For instance, PLLaVA [1] adopts the 2D Adaptive Average pooling to reduce the visual tokens.

2. The experimental analysis is not comprehensive enough, leading to the results being hard to understand. For example, in Table 1,  C-Abstractor and DeCo have the same number of visual tokens fed into LLM, and C-Abstractor performs local enhancement on the compressed tokens. It seems that C-Abstractor should outperform DeCo, but the experimental results show that DeCo performs better. The reasons need to be further explained.

3. Lack of necessary experiments. All experiments in DeCo are performed at normal resolution, but the visual token compression is useful for high-resolution scenes. Unfortunately, this part of the experiment is not reflected in the paper. Authors are suggested to add this experiment.

[1] Xu L, Zhao Y, Zhou D, et al. Pllava: Parameter-free llava extension from images to videos for video dense captioning[J]. arXiv preprint arXiv:2404.16994, 2024.

**Questions:**

1. Although 2D Adaptive Average pooling can reduce the computational burden, the MLLM computation is mainly in the LLM. The increase in the computation of the projector does not have a significant impact on the overall computation. Therefore, from this point of view, does the module with learnable parameters such as convolution achieve better performance than 2D Adaptive Average pooling?

2. Table 3 shows that DeCo's performance is very close to MLP's performance, but the number of tokens is much lower. The reasons why DeCo can outperform MLP require further explanation.

3. Writing typo. L215 "an MLLM" -> "a MLLM".

**Post-rebuttal**

Unfortunately, the authors don't provide detailed responses to my concerns. Therefore, I have decided to give the final recommendation: **Strong reject**. After reading the comments from other reviewers, I conclude the reasons for the rejection decision.

1. We achieve the agreement that the novelty of this paper is limited (Reviewer NfRX, nzvd, and YoWL). Average pooling is a common practice in modern MLLM. This paper is just an analysis of its effectiveness, which does not mean so much to the ICLR community.

2. Limited experimental analysis. This paper lacks a detailed experimental analysis of why other visual projectors perform worse than average pooling (Reviewer NfRX and YoWL). Also, the MLLM benchmarks are limited (Reviewer prwT). The existing problems make the paper unable to validate the proposed method.

3. The proposed R-GAE is close-source, underweighting the significance of R-GAE.

Besides, **it seems that the authors have given up this paper, which is not a good example for open discussion in ICLR**. Overall, I agree with other reviewers that **this paper should be rejected**.

---

### Official Review · Reviewer_nzvd · 2024-10-27

**Soundness:** 4
**Presentation:** 3
**Contribution:** 3
**Rating:** 6
**Confidence:** 4

**Summary:**

The manuscript conducts a thorough analysis of existing visual projectors and based on the proposed R-GAE, reveals that compressive projectors (e.g., QFormer) experience a "double abstraction" problem. To address this issue, the authors propose "Decouple Token Compression from Semantic Abstraction (DeCo)."

**Strengths:**

1. The analysis of existing visual projectors is convincing.
2. A simple 2D Adaptive Pooling outperforms traditional compressive projectors in both performance and efficiency.

**Weaknesses:**

1. The novelty of this paper is limited; the proposed method simply uses 2D adaptive average pooling, which may result in loss of detailed information. Also, the “double abstraction” problem is not solved; the pooling operation is also an abstraction in the vision modality.
2. In the manuscript, Q-Former, C(D)-Abstractor, and DeCo all use 144 visual tokens. If the number of visual tokens increases (reduce the compression rate), will DeCo still outperform other methods?

**Questions:**

please refer to weaknesses.

---

### Official Review · Reviewer_prwT · 2024-10-30

**Soundness:** 3
**Presentation:** 3
**Contribution:** 3
**Rating:** 3
**Confidence:** 5

**Summary:**

The authors first reveal the drawbacks of the previous comprehension-based projector such as Q-Former. The authors use the R-GAE explainability tool to reveal limited visual semantics and repetitive patterns. Then the authors propose a new token comprehension technique based on adaptive average pooling. The authors conduct a series of experimental results to demonstrate the effectiveness of the proposed module.

**Strengths:**

1. The authors present an analysis of the relevancy between compressed tokens and input images.

2. The proposed method (adaptive average pooling comprehension) is easy to follow.

**Weaknesses:**

1. About the selected benchmarks.

    The authors claim that 'The non-compressed projector' struggles with high-resolution or video benchmarks. However, the selected benchmarks in this paper (e.g., SEED, MME) do not belong to these two categories. Thus, the non-compressed projector performs best in Table 1, and can not demonstrate the necessity of using the compressed projector. I suggest the author provides more experimental results on (1) high-resolution benchmarks, such as InfographicVQA; and (2) video benchmarks, such as Video-MME. And the authors may consider adding the time-efficiency comparison.

2. Limited MLLM baselines

    This paper is based on LLaVA-1.5 merely, which is far behind current SOTA models. I suggest the authors provide the experimental results on more recent MLLM models.

3. More related works such as token reduction

    Instead of designing a compressive projector, another direction is using the non-compressive projector plus the token reduction techniques, such as [1,2]. However, this paper does not discuss these related works.
    - [1] An Image is Worth 1/2 Tokens After Layer 2: Plug-and-Play Inference Acceleration for Large Vision-Language Models, ECCV 2024
    - [2] Not All Patches are What You Need: Expediting Vision Transformers via Token Reorganizations, ICLR 2022

**Questions:**

1. What's the language instruction used in Figure 3?

---

### Official Review · Reviewer_NfRX · 2024-11-04

**Soundness:** 2
**Presentation:** 2
**Contribution:** 2
**Rating:** 3
**Confidence:** 3

**Summary:**

This paper use avg pooling to reduce the number of vision tokens.

**Strengths:**

1. The writing is clear. Reader know the contributionL use avg pooling to reduce the number of vision tokens.

**Weaknesses:**

1. Contribution not enough as a ICLR paper. Average pooling has been using in a lot of other papers, including Phi-3.5-Vision, Qwen2-VL, etc. Not a new technique.
2. For all content up to page 5, all observations with Q-former are visualization. Picked qualitative study should not and can not lead to the solid conclusion of why Q-former fails.

**Questions:**

1. There is still no solid conclusion/reasoning w.r.t. why convolution based approach like C-abstractor fails.

---

### Note · Authors · 2024-12-03

**Comment:**

We sincerely thank all reviewers for your valuable feedback and the time you dedicated to reviewing our work. **We have thoroughly read and carefully considered all of your suggestions.**

After careful consideration, we acknowledge that the organizational structure of the DeCo paper has some issues. The primary goal of DeCo is to explore which projectors are effective for MLLM from an explainable perspective, rather than to propose the Average Pooling design. Based on the feedback from reviewers, we will:
1) Enhance the analysis section and include quantitative experiments for a more comprehensive evaluation of existing projectors beyond performance.
2) Improve the experimental section, particularly regarding high-resolution and multi-frame video scenarios.

Thank you again to all reviewers for your constructive feedback.

**Withdrawal Confirmation:**

I have read and agree with the venue's withdrawal policy on behalf of myself and my co-authors.